# Mild Acquired von Willebrand Syndrome and Cholestasis in Pediatric and Adult Patients with Fontan Circulation

**DOI:** 10.3390/jcm12031240

**Published:** 2023-02-03

**Authors:** Katharina Meinel, Felicitas Korak, Martin Dusleag, Tanja Strini, Daniela Baumgartner, Ante Burmas, Hannes Sallmon, Barbara Zieger, Axel Schlagenhauf, Martin Koestenberger

**Affiliations:** 1Department of Pediatrics and Adolescent Medicine, Division of Pediatric Cardiology, Medical University of Graz, Auenbruggerplatz 34/2, 8036 Graz, Austria; 2Department of Pediatrics and Adolescent Medicine, Division of General Pediatrics and Adolescent Medicine, Medical University of Graz, Auenbruggerplatz 34/2, 8036 Graz, Austria; 3Department of Congenital Heart Disease/Pediatric Cardiology, Deutsches Herzzentrum der Charité (DHZC), Augustenburgerplatz 1, 13353 Berlin, Germany; 4Department of Pediatrics and Adolescent Medicine, Division of Pediatric Hematology and Oncology, Faculty of Medicine, Medical Center—University of Freiburg, 79098 Freiburg, Germany

**Keywords:** Fontan circulation, Fontan-associated liver disease, cholestasis, bile acids, hemostasis, thromboprophylaxis, acquired von Willebrand syndrome

## Abstract

**Background:** Hemodynamic alterations in Fontan patients (FP) are associated with hemostatic dysbalance and Fontan-associated liver disease. Studies of other hepatopathologies indicate an interplay between cholestasis, tissue factor (TF), and von Willebrand factor (VWF). Hence, we hypothesized a relationship between the accumulation of bile acids (BA) and these hemostatic factors in FP. **Methods:** We included 34 FP (Phenprocoumon *n* = 15, acetylsalicylic acid (ASA) *n* = 16). BA were assessed by mass spectrometry. TF activity and VWF antigen (VWF:Ag) were determined by chromogenic assays. VWF collagen-binding activity (VWF:CB) was assessed via ELISA. **Results:** Cholestasis was observed in 6/34 FP (total BA ≥ 10 µM). BA levels and TF activity did not correlate (*p* = 0.724). Cholestatic FP had lower platelet counts (*p* = 0.013) from which 5/6 FP were not treated with ASA. VWF:Ag levels were increased in 9/34 FP and significantly lower in FP receiving ASA (*p* = 0.044). Acquired von Willebrand syndrome (AVWS) was observed in 10/34-FP, with a higher incidence in cholestatic FP (4/6) (*p* = 0.048). **Conclusions:** Cholestasis is unexpectedly infrequent in FP and seems to be less frequent under ASA therapy. Therefore, ASA may reduce the risk of advanced liver fibrosis. FP should be screened for AVWS to avoid bleeding events, especially in cholestatic states.

## 1. Introduction

The Fontan operation is a palliative procedure in patients with functionally univentricular hearts, which surgically redirects systemic venous blood return to the pulmonary circulation in the absence of a subpulmonary ventricle [1]. Multiple factors are contributing to an increased risk of thromboembolic events in patients with Fontan circulation which are related to Virchow’s triad of thrombogenesis comprising altered blood flow, endothelial injury, and hypercoagulability [2,3]. First of all, the loss of pulsatility and subsequent elevated systemic venous pressure predisposes to thrombus formation through stasis of blood [3]. Additionally, a diminished cardiac output in Fontan patients was reported to increase the risk for thrombosis [4]. Finally, artificial intravascular material, chronic hypoxia, and surgical manipulation are thought to be involved in endothelial injury [5,6]. Hemodynamic changes and endothelial dysfunction were shown to increase levels of von Willebrand factor (VWF), platelet reactivity, and thrombin generation in the Fontan circulation [7,8]. However, these aforementioned defects could also be associated with VWF degradation leading to acquired von Willebrand syndrome (AVWS), but data on VWF function in Fontan patients is lacking.

In addition to coagulation abnormalities, the increased venous pressure in the Fontan circulation is also leading to a chronic state of hepatic congestion. The non-inflammatory structural and functional hepatic changes associated with hemodynamic alterations of the Fontan circulation are summarized as Fontan-associated liver disease (FALD) [9,10]. As FALD progresses, sinusoidal fibrosis becomes inevitable [9]. Intrahepatic coagulation was shown to foster fibrosis in Fontan patients via thrombin generation and intraparenchymal fibrin deposition [11]. Subsequent platelet activation was proven to promote liver fibrosis by the activation of hepatic stellate cells in a mouse model [12]. Additionally, mechanistic and clinical observations in other hepatopathies suggest a link between changes in VWF and liver injury [13]. Taken together, intrahepatic coagulation processes play a detrimental role in the progression of FALD. However, the actual hemostatic trigger for these processes is unknown.

Elevated bile acid (BA) levels are hypothesized to play a key role in the decryption of human tissue factor (TF), which in complex with activated coagulation factor VII (FVIIa) serves as a hemostatic trigger [14]. In a previous study, we could demonstrate increased hepatic TF activity upon incubating hepatocytes with certain BA in vitro [14]. As Fontan patients tend to exhibit mild cholestasis by hepatic congestion, elevated BA levels may subsequently increase TF activity as the principal trigger of plasmatic coagulation. Hence, we hypothesized an interplay between cholestasis and a hemodynamically induced procoagulant state fostering FALD in patients with Fontan circulation.

## 2. Material and Methods

**Study design and patients’ characteristics:** We conducted an observational study at the Division of Pediatric Cardiology and the GUCH (grown-up with congenital heart disease) Unit of the Medical University of Graz. We included 34 Fontan patients (14 females, 20 males) aged 5 to 38 years from whom we collected blood samples between May 2020 and May 2021. Non-fasting serum samples were taken for BA measurements from all study participants (*n* = 34). A total BA (tBA) cut-off value of ≥10 µM was set to determine cholestasis within the study cohort. Citrate plasma was available in 33/34 Fontan patients (14 females, 19 males) for coagulation analysis. From the 33 available plasma samples, 2 patients were not subjected to any anticoagulant medication during sample collection. The remaining subcohorts comprised 15 patients, receiving vitamin K antagonist Phenprocoumon (PhC) and 16 patients medicated with platelet aggregation inhibitor acetylsalicylic acid (ASA) (Appendix A).

**Blood samples:** Blood samples from children and adolescent Fontan patients were collected via venipuncture at the Division of Pediatric Cardiology of the Medical University of Graz during routine diagnostic workup. Blood samples from adult Fontan patients were obtained by venipuncture at the GUCH Unit of the Medical University of Graz. Serum samples (*n* = 34) were collected in serum tubes (1.4 mL) which were centrifuged at 2000× *g* for 10 min within 3 h before storage at −80 °C until analysis. Citrate plasma samples (*n* = 33) were collected using a single citrate tube (3 mL), which were processed to platelet-poor plasma by centrifugation at 2598× *g* for 10 min within 3 h and stored at −80 °C until analysis.

**BA analysis:** BA levels including unconjugated, taurine-, and glycine-conjugated BA species (Appendix A) were measured by high-performance liquid chromatography (HPLC) combined with tandem mass spectrometry (MS/MS) as described previously [15]. Briefly, plasma samples were prepared after the protocol of Humbert et al. [16]. After the addition of internal standards d4-DCA, d4-LCA, d4-GLCA, d4-GCDCA, and d4-TDCA, 0.2 nmol each, plasma samples (10 μL) were vortexed for one minute. Then, 400 µL of acetonitrile (80% *v*/*v*; Sigma Aldrich, Taufkirchen, Germany) was added for deproteination. After vortexing, the precipitate was removed by centrifugation at 3200× *g* for 12 min. The supernatant was dried under a stream of nitrogen (40 °C). The samples were re-dissolved in 100 µL of mobile phase B (methanol with 1.2% *v*/*v* formic acid and 0.38% *w/v* ammonium acetate) and transferred to an autosampler. Individual BA were separated by HPLC using a reversed-phase C18 column (Macherey-Nagel, Düren, Germany) and a kinetex pentafluorophenyl column (Phenomenex, Aschaffenburg, Germany). Quantification and characterization were achieved using a Q Exactive™ mass spectrometer (Thermo Fisher Scientific, Waltham, MA, USA) and a high-performance quadrupole precursor selection with high-resolution and accurate-mass (HR/AM) Orbitrap™ detection [16]. In our Fontan patients, cholestasis was defined by a cut-off value of tBA ≥ 10 µM. The cut-off was established during validation of the HPLC-MS method in our laboratory with a broad range of samples from patients that were externally diagnosed with varying degrees or absence of cholestasis.

**TF activity analysis:** The procoagulant activity of human TF in plasma samples was determined through a commercial chromogenic microplate assay (BioMedica Diagnostics, Windsor, CA, USA). The assay, which is based on the conversion of factor X to factor Xa followed by the enzymatic cleavage of a chromogenic substrate, was performed according to the manufacturer’s instructions.

**Von Willebrand factor analysis:** VWF antigen (VWF:Ag) and VWF collagen binding capacity (VWF:CB) were determined, as described previously [17]. Briefly, VWF:Ag was measured in sodium citrate plasma using an in-house ELISA (Sutor Semin Thromb Haemost 2001). Collagen type I was immobilized on a microtiter plate, and VWF:CB in plasma was measured photometrically via the ELISA technique. Ratios of VWF:CB/VWF:Ag were calculated reflecting the biological capacity of the available VWF to bind to collagen. Ratios < 0.7 were considered pathological and indicative of acquired von Willebrand syndrome (AVWS). VWF was determined using appropriate primary and secondary antibodies and 3.30-diaminobenzidine/cobalt chloride. Standard human plasma was used as control. AVWS was diagnosed if the VWF:CB/VWF:Ag-ratio was reduced.

**Endothelial damage analysis:** The tissue-type plasminogen activator (tPA) antigen level in plasma was assayed using a commercial in vitro ELISA kit (Hyphen BioMed, Neuville-sur-Oise, France). A commercial in vitro ELISA kit (Hyphen BioMed) was performed to measure the plasminogen activator inhibitor-1 (PAI-1) antigen concentration. The thrombomodulin ™ levels in patient plasma were determined using a commercial ELISA test kit (Abcam, Cambridge, UK). The test procedures were conducted according to the provided protocol.

**Clinical data:** We collected clinical data and routine laboratory data of the included Fontan patients which comprised current medication, serum bilirubin, gamma-glutamyl transferase (GGT), as well as prothrombin time (PT), and platelet count, amongst others.

**Ethics:** This clinical study has been approved by the Austrian ethics committee (EK-Nr.32-376ex19/20) and was performed in accordance with the ethics standards as laid down in the 1964 Declaration of Helsinki and its later amendments or comparable ethical standards. Informed parental consent/informed consent was obtained for each included subject.

**Statistical analysis:** The statistical analysis was performed in IBM SPSS Statistics 28.0.0.0 and GraphPad Prism software. The changes in the examined parameters were visualized using GraphPad Prism Software. After testing for normal distribution, comparison of data from cholestatic and non-cholestatic Fontan patients was made via a Man-*n*–Whitney U-test. Spearman’s correlation coefficient and corresponding *p*-values were used to investigate the relationships between the examined parameters. The statistical analysis regarding PT was limited to the patient subcohort receiving PhC.

## 3. Results

### 3.1. Demographic Data and Clinical Characteristics

Based on a tBA cut-off value of ≥10 µM, cholestasis was identified in 6/34 Fontan patients, equaling 17.7% of all patients included in the study. Concerning standard laboratory markers indicating hepatic injury, cholestatic Fontan patients had significantly higher bilirubin and GGT levels compared to non-cholestatic Fontan patients (Table 1).

### 3.2. Bile Acids

The tBA concentrations of the cholestatic Fontan subcohort ranged from 10.85 to 59.01 µM (median: 48.38 µM, IQR: 28.42–53.49), whereas the tBA levels of non-cholestatic Fontan patients ranged from 0.72 to 7.86 µM (median: 3.31 µM, IQR: 2.17–5.10). The tBA levels were significantly higher in Fontan patients ≥18- than in patients <18 years of age (*p* = 0.033) (Figure 1A). BA profiles of the cholestatic and non-cholestatic Fontan groups were established by determining the mean of the relative fraction for each of the 15 common human BA (Figure 1B). The relative fractions of glycocholic acid (GCA), taurocholic acid (TCA), taurolitocholic acid (TLCA), and tauroursodeoxycholic acid (TUDCA) were significantly higher, whereas the relative fraction of ursodeoxycholic acid (UDCA) was significantly lower in cholestatic Fontan patients compared to non-cholestatic patients (Appendix A).

### 3.3. Laboratory Parameters and Clinical Data

In PhC-treated Fontan patients with cholestasis, the PT was significantly lower (median: 25.0, IQR: 21.8–28.3) compared to PhC-treated study participants without cholestasis (median: 33.0, IQR: 30.0–41.5, *p* = 0.013). Most cholestatic Fontan patients (5/6) were not treated with ASA (Appendix A) and showed significantly lower platelet counts than non-cholestatic study participants (*p* = 0.013) (Table 1). In non-cholestatic individuals, platelet counts (×10^3^/µL) were not affected by the type of antithrombotic treatment, as they were comparable between ASA- (median: 167.0, IQR: 132.5–215.8) and PhC-treated (median: 152.0, IQR: 113.0–222.0) patients (*p* = 0.677).

### 3.4. Tissue Factor Activity Analysis

The TF activity was not significantly higher in cholestatic Fontan patients (median: 6.88 pM, IQR: 3.25–13.71) compared to non-cholestatic patients (median: 7.73 pM, IQR: 2.65–16.43, *p* = 1.000).

### 3.5. Markers of Endothelial Damage

The endothelial damage was further evaluated by comparison of the TM levels (ng/mL) of cholestatic (median: 4.119, IQR: 2.977–5.836) and non-cholestatic Fontan patients (median: 5.416, IQR: 3.762–6.597) but without a statistically significant difference (*p* = 0.276). Likewise, both endothelial markers tPA and PAI-1 (ng/mL) showed no statistically significant differences between the subcohorts (tPA cholestasis median: 2.78, IQR: 2.63–5.10; tPA no cholestasis median: 2.76, IQR: 2.21–4.29, *p* = 0.633; PAI-1 cholestasis median: 9.51, IQR: 7.47–12.81; PAI-1 no cholestasis median: 10.47, IQR: 5.66–22.50, *p* = 0.760).

### 3.6. VWF Analysis

The VWF:Ag levels were compared between cholestatic and non-cholestatic Fontan patients. In both groups, VWF:Ag levels were generally elevated (cholestatic median 146.7%, IQR: 125.1–164.7; non-cholestatic median: 138.8%, IQR: 119.6–176.4), however, without a statistically significant difference between the groups (*p* = 0.701) (Figure 2A). Interestingly, VWF:Ag levels were significantly lower in Fontan patients receiving ASA (median: 109.0%, IQR: 95.0–122.5) than in Fontan patients without ASA treatment (median: 125.0%; IQR: 111.5–171.5, *p* = 0.0436) (Figure 2B). VWF:CB was comparable in cholestatic- (median: 84.0%, IQR: 65.3–144.0) compared to non-cholestatic Fontan patients (median: 92.5%, IQR: 77.0–144.0, *p* = 0.938) (Figure 2A).

However, the VWF:CB/VWF:Ag ratio was pathologically low, indicating AVWS in 10/34 Fontan patients (29.4%). Although the VWF:CB/VWF:Ag ratio did not differ statistically significantly between cholestatic (median: 0.70, IQR: 0.56–0.78) and non-cholestatic Fontan patients (median: 0.77, IQR: 0.73–0.92, *p* = 0.112) (Figure 3), ratios below the pathological threshold were significantly more frequent in cholestatic (4/6) than in non-cholestatic Fontan patients (6/28) (*p* = 0.048) (Figure 3).

Fontan patients with decreased VWF:CB/VWF:Ag ratio ≤ 0.7 were compared with Fontan patients with VWF:CB/VWF:Ag ratio > 0.7 concerning invasively measured hemodynamic parameters and NTproBNP levels. We found no statistically significant differences for the mean Fontan pathway pressure (ratio ≤ 0.7 median: 12.5 mmHg, IQR 10.2–14.5; ratio > 0.7 median: 11.0 mmHg, IQR: 10.0–12.0, *p* = 0.111), pulmonary capillary wedge pressure (ratio ≤ 0.7 median: 7.0 mmHg, IQR 5.0–9.0; ratio > 0.7 median: 6.0 mmHg, IQR: 5.0–8.0, *p* = 0.419) or transpulmonary pressure gradient (ratio ≤ 0.7 median: 5.0 mmHg, IQR 4.0–6.8; ratio > 0.7 median: 5.0 mmHg, IQR: 4.0–6.0, *p* = 0.592). Moreover, there was no statistically significant difference in NTproBNP levels (ratio ≤0.7 median: 200 ng/mL, IQR 83–338; ratio > 0.7 median: 148 ng/mL, IQR: 82–310, *p* = 0.655).

### 3.7. Correlation Analysis

A Spearman’s rank correlation was performed to determine a potential relationship between BA parameters and TF activity and endothelial damage parameters. No significant correlation was determined between the analyzed BA variables and TF activity (*p* ≥ 0.05). Furthermore, the BA parameters did not correlate with the endothelial damage markers TM, tPA, PAI-1, VWF:Ag, and VWF:CB (*p* ≥ 0.05).

## 4. Discussion

FALD is defined as a hepatic disorder comprising structural and functional alterations which are arising from hemodynamic changes and chronic systemic venous congestion following Fontan surgery [9]. Albeit liver fibrosis seems to be universally present in Fontan patients, hepatic laboratory markers are usually not reflecting histopathological findings [18,19,20,21]. In advance of assessing the tBA values within our study cohort, a high incidence rate of cholestasis was hypothesized. However, the HPLC-MS/MS analysis revealed elevated tBA levels (≥10 µM) only in six out of 34 Fontan patients. Upon comparing the BA pools of the cholestatic and non-cholestatic Fontan subcohorts, the cholestatic subjects showed a statistically significant increase in GCA, TCA, TLCA and TUDCA. In contrast to the aforementioned, UDCA was significantly decreased in Fontan patients with cholestasis. Out of six cholestatic Fontan patients, five were ≥18 years of age, which is in accordance with previous research findings describing the time since Fontan completion as the main risk factor for FALD development [9,22]. In accordance with mild elevations of biochemical hepatic parameters being a common secondary sequela in Fontan patients, serum bilirubin and GGT values were compared in the study cohort. As a result, increased serum bilirubin and GGT levels were discovered in the cholestatic subcohort, suggesting that hepatic congestion serves as a potential confounder of the heightened BA and hepatic function markers bilirubin and GGT.

Even though the exact pathophysiology is not completely understood, a relationship between the advance of FALD and intrahepatic thrombosis has been assumed [8,9,23]. As increased hepatic TF activity was observed in the setting of elevated bile acids, we analyzed the TF activity as the principal trigger of plasmatic coagulation in Fontan patients [14]. In our study population, however, the plasmatic analysis of TF activity in cholestatic and non-cholestatic Fontan patients revealed no significant difference between both groups. Additionally, no correlation was found between the BA and TF activity of the examined patients, arguing against a BA-mediated decryption of hepatic TF in cholestatic patients with Fontan circulation. This mechanism is most likely relevant in primary obstructive liver diseases where bile acid accumulation precedes the activation of coagulation. However, patients with Fontan circulation show coagulatory abnormalities from the beginning due to altered hemodynamics, and only a fraction of patients develop cholestasis years after Fontan completion. Hence, intrahepatic coagulation activation most likely fosters liver fibrosis and precedes the obstruction of bile flow.

Nevertheless, multiple factors are contributing to an increased risk for thromboembolic events in the Fontan circulation including disturbed blood flow, endothelial injury, and coagulopathy [3]. However, controversies exist regarding optimal thromboprophylaxis in Fontan patients [24]. To date, the superiority of either anticoagulation or antiplatelet therapy (ASA) has not been proven. Therefore, treatment recommendation for each patient was based on subjective thromboembolic risk assessment of the treating physician. In two independent randomized controlled trials comparing vitamin K antagonists (e.g., PhC) and ASA, no significant differences have been demonstrated regarding thromboembolic events in children with Fontan circulation [25,26]. Within all included study participants, 16 patients were treated with ASA (cholestatic *n* = 1, non-cholestatic *n* = 15), whereas 15 patients received PhC (cholestatic *n* = 4, non-cholestatic *n* = 11). Generally, we are preferring the prescription of ASA for thrombophylaxis in our pediatric patients due to the higher risk of bleeding in case of trauma. At the appearance of clear indications such as atrial arrythmias or thromboembolic events, amongst others, pediatric and adult patients are switched to anticoagulation with PhC. Moreover, life events such as pregnancy and additional surgery of course might require changes to thrombophylaxis strategies. In our study, PhC-treated cholestatic Fontan patients had a significantly shorter PT than non-cholestatic patients with PhC, hinting at an impaired synthetic function of the liver, although a variance in PhC treatment cannot be entirely excluded. Independent of PhC treatment, however, platelet counts were significantly lower in cholestatic Fontan patients whereby five out of six were not treated with ASA. In non-cholestatic Fontan patients, platelet counts were comparable between PhC- and ASA-treated individuals. These observations are either pointing to a decreased platelet biogenesis or an increased platelet activation and consumption in cholestatic Fontan patients. In a mouse model, platelet activation has been shown to promote liver fibrosis by activating hepatic stellate cells [12]. Furthermore, the daily use of aspirin was associated with less severe histologic findings and a lower risk of advanced fibrosis progression over time in non-alcoholic fatty liver disease [27]. In Fontan patients, ASA therapy might reduce the risk for advanced liver fibrosis and therefore the severity of FALD; however, this needs further investigation by larger clinical trials.

Endothelial function and fibrinolysis have been investigated previously as potential contributors to thrombosis in Fontan patients. Binotto et al. reported increased VWF levels, pointing to a dysfunction of the vascular endothelium [6]. This is in accordance with our findings of enhanced VWF:Ag levels in all our Fontan patients indicating either endothelial perturbation or increased platelet activation. However, we could not find a significant difference between cholestatic and non-cholestatic Fontan patients.

AVWS is characterized by the loss of high molecular weight multimers of VWF leading to an impaired VWF function, which can be either shear stress-induced or associated with increased platelet activation [28]. It has been shown that the combined interpretation of VWF:Ag, VWF:CB, and VWF:CB/VWF:Ag ratio increases the sensitivity of diagnosing AVWS [29]. A low VWF:CB/VWF:Ag ratio < 0.7 has been shown to correlate with the loss of high molecular weight multimers [29]. Generally, the VWF:CB/VWF:Ag ratio was decreased in 29.4% of all included Fontan patients. Moreover, the incidence of pathological VWF:CB/VWF:Ag ratio was higher in cholestatic Fontan patients. One explanation for our findings could be an increased release of VWF by the dysfunctional vascular endothelium of Fontan patients leading to generally enhanced levels of detectable VWF:Ag. Alternatively, the altered hemodynamics in the Fontan circulation might change the three-dimensional VWF structure and enhance the proteolysis of VWF by ADAMTS-13. Similarly, a loss of high molecular weight multimers of VWF has been reported in patients after continuous-flow left ventricular assist device implantation [30]. However, it remains unclear why all Fontan patients are not affected by a mild AVWS despite comparable invasively measured hemodynamic parameters.

Lastly, we investigated fibrinolysis via the release of tPA and PAI-1, as the processes are primarily mediated by the vascular endothelium. In our study, the analyzed endothelial parameters did not differ significantly between cholestatic- and non-cholestatic Fontan patients. Likewise, in the study of Binotto et al., tPA and PAI-1 levels and activity did not deviate significantly from the control group, showing that not all endothelial functions are impaired following the Fontan procedure [6].

We acknowledge that the representativeness of our results may be limited due to the small number of analyzed patients per study group. Statistical evaluation was constrained by the low incidence rate of cholestasis within the study cohort. Furthermore, as 4/6 cholestatic Fontan patients received PhC, the hemostatic statistical evaluation was limited to the PhC subcohort. Still, we believe that the results of our study revealed valuable new insights into the pathophysiology of FALD with inherent clinical impact as clinical trials investigating coagulation in Fontan patients are sparse.

## 5. Conclusions

Firstly, the low incidence rate of cholestasis in Fontan patients represents a novel clinical finding. However, elevated bile acid levels were not associated with increased tissue factor activity or endothelial dysfunction. Secondly, the absence of cholestasis in patients receiving ASA is striking and warrants further investigation. Thirdly, the Fontan circulation is associated with a mild acquired von Willebrand syndrome. However, ongoing clinical trials are required to investigate the risk of bleeding events in Fontan patients with acquired von Willebrand syndrome. Due to the higher incidence and possibly impaired hepatic synthetic function in cholestatic patients, testing for acquired von Willebrand syndrome should be considered prior to surgery to avoid bleeding events.

## Figures and Tables

**Figure 1 jcm-12-01240-f001:**
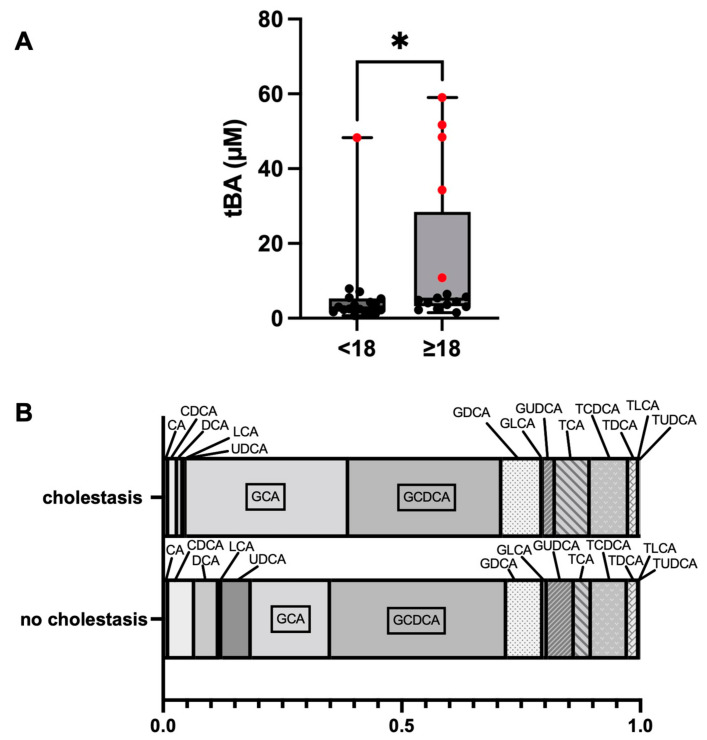
Comparison of the distribution of total bile acid (tBA) concentrations between Fontan patients <18 and ≥18 years of age. Cholestatic patients are highlighted by the red dots (**A**). BA profiles of cholestatic and non-cholestatic Fontan patients (**B**). * *p* ≤ 0.05. **Abbreviations:** See Appendix A.

**Figure 2 jcm-12-01240-f002:**
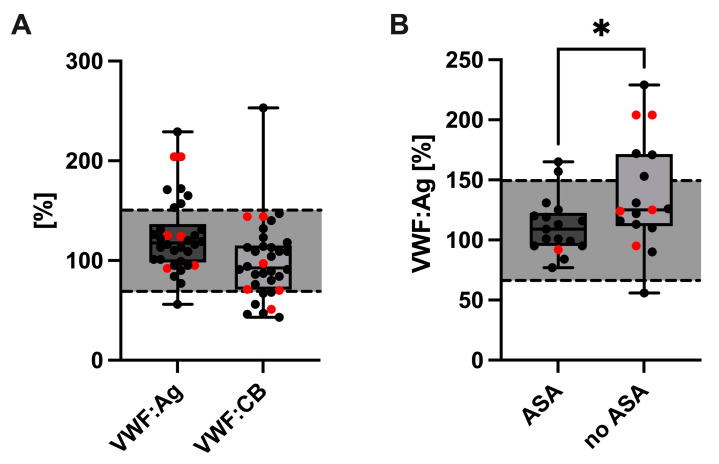
VWF:Ag and VWF:CB (**A**) values in cholestatic (**red dots**) and non-cholestatic Fontan patients. Comparison of VWF:Ag levels between Fontan patients with and without acetylsalicylic acid (ASA) treatment. Cholestatic patients are highlighted by the red dots (**B**). * *p* ≤ 0.05. **Abbreviations:** VWF, von Willebrand factor; VWF:Ag, VWF antigen; VWF:CB, VWF collagen binding activity.

**Figure 3 jcm-12-01240-f003:**
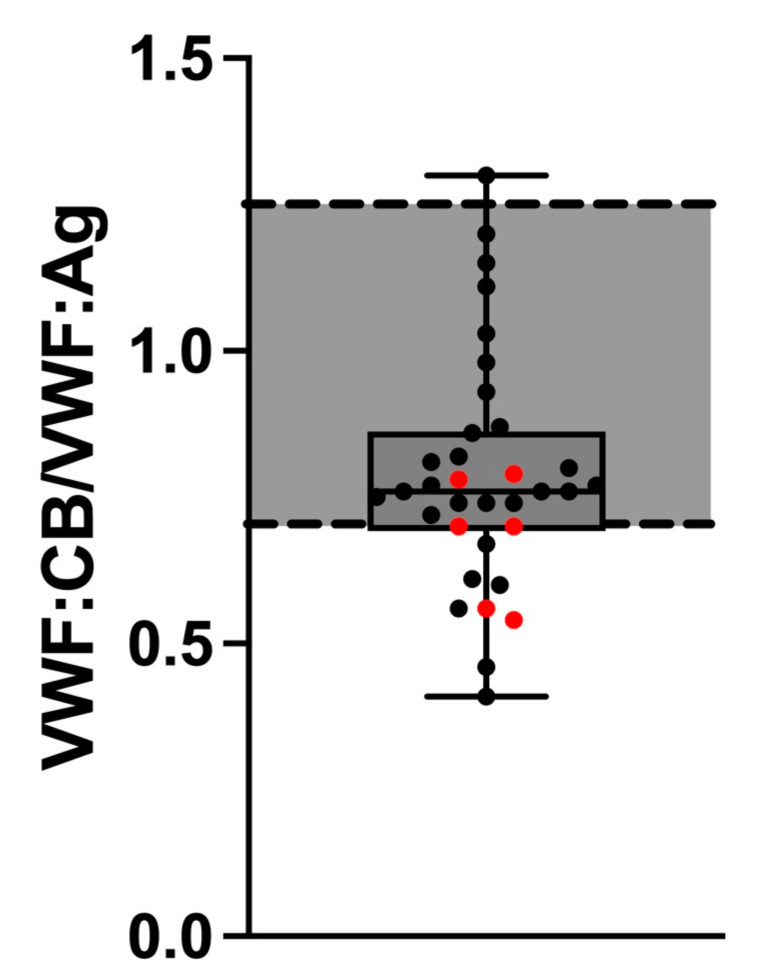
VWF:CB/VWF:Ag ratio in cholestatic (**red dots**) and non-cholestatic Fontan patients. **Abbreviations:** VWF:Ag, von Willebrand factor antigen; VWF:CB, von Willebrand factor collagen binding activity.

**Table 1 jcm-12-01240-t001:** Characteristics of cholestatic and non-cholestatic pediatric and adult Fontan patients.

	Non-Cholestatic FP (*n* = 28)	Cholestatic FP (*n* = 6)	*p*-Value
**Baseline demographic data**			
**Age at Fontan Procedure** (y)	3.6 (2.6–6.5)	3.4 (3.0–4.0)	0.973
**Gender, female** (*n*,%)	13 (46.4)	1 (16.7)	0.179
**Dominant ventricle:** (*n*)			0.383
Left	11	4	
Right	13	1	
Both	4	1	
**Type of Fontan:**			0.724
TCPC (lateral tunnel)	8	1	
ECFO	19	5	
Other (Kawashima)	1	0	
**Last follow-up**			
**Age at follow-up** (y)	23.5 (16.0–32.3)	12.4 (9.3–28.0)	0.183
**Time since Fontan** (y)	18.0 (13.5–28.5)	9.5 (6.3–24.7)	0.191
**Ejection fraction:** (*n*)			0.638
>50%	27 (96.5%)	6 (100%)	
<50%	1 (3.5%)	0	
**AV-valve incompetence:** (*n*)			0.401
Mild/moderate	25 (89.3%)	6 (100%)	
Severe	3 (10.7%)	0	
**Erythrocytes** (×10^6^/µL)	5.2 (4.7–5.6)	5.5 (5.3–5.7)	0.175
**Hemoglobin** (g/dL)	15.0 (13.4–15.4)	16.4 (15.7–17.0)	0.066
**Hematocrit** (%)	43.5 (40.0–45.0)	46.5 (45.0–48.7)	**0.044**
**Platelet count** (×10^3^/µL)	196.0 (152.0–242.0)	116.0 (70.3–217.0)	**0.013**
**ALT** (U/L)	25.5 (19.0–30.0)	26.0 (21.8–43.0)	0.428
**AST** (U/L)	28.5 (22.0–36.0)	30.5 (25.8–42.3)	0.485
**GGT** (U/L)	46.5 (36.5–73.8)	99.5 (60.8–198.5)	**0.032**
**Cholinesterase** (U/L)	7241 (6263–8537)	6187 (5814–7687)	0.243
**Lactate dehydrogenase** (U/L)	227.5 (175.0–255.8)	216.5 (167.5–278.8)	0.935
**Bilirubin** (mg/dL)	0.8 (0.6–1.2)	1.9 (1.2–2.9)	**0.003**
**Creatinine** (mg/dL)	0.6 (0.5–0.8)	0.8 (0.7–0.9)	0.219
**Total protein** (g/dL)	7.5 (6.9–7.6)	7.4 (7.2–7.7)	0.785
**Albumin** (g/dL)	4.7 (4.5–4.9)	4.7 (4.3–4.8)	0.629
**NT-ProBNP** (ng/mL)	149.5 (73.8–269.3)	338.0 (103.8–396.3)	0.179
**Antithrombin III** (%)	101.0 (94.0–106.5)	85.0 (80.5–91.0)	**0.002**
**Medication:** (*n*)			0.204
Acetylsalicylic acid (ASA)	16	1	
Phenprocoumone (PhC)	11	4	
ACE inhibitor	15	2	
Antiarrhythmic	5	3	
PH medication	2	2	

**Significant differences are shown in bold. Abbreviations:** ALT, alanine aminotransferase; AST, aspartate aminotransferase; AV, atrioventricular; ECFO, extracardiac Fontan operation; FP, Fontan patients; GGT, gamma-glutamyl transferase; NTproBNP, N-terminal pro-B-type natriuretic peptide; PH, pulmonary hypertension; TCPC, total cavopulmonary connection; y, years.

## Data Availability

The data presented in this study are available on request from the corresponding author. The data are not publicly available due to ethical reasons.

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
