# Peer review of "Mild Acquired von Willebrand Syndrome and Cholestasis in Pediatric and Adult Patients with Fontan Circulation"

_jcm, 2023, doi:10.3390/jcm12031240_

Round 1
Reviewer 1 Report
I read the manuscript “Mild acquired von Willebrand syndrome and cholestasis in pe-1 diatric and adult patients with Fontan circulation” with great interest.
Here are my comments:
1. Manuscript needs to be extensively revised for English and grammatical errors
2. Where did the cut-off of 10micromole for bile acids come from?
3. How was the determination made between prescribing patients PhC vs. ASA?
4. Was this a truly prospective study? Meaning were the patients enrolled in the study at the time when they underwent their Fontan with active consent about this specific study? If not, this cannot be considered a prospective study
5. For TCPC, what was the approach? Lateral tunnel or atrial Fontans?
6. Does the ventricular morphology in table 1 mean the dominant ventricle?
7. Figure 1a and 2 seem to be confusing. It may be better to place a proper Box plot instead of scattered dots.
8. I do not think the conclusion of “ASA intake may reduce the risk 310 of advanced liver fibrosis in Fontan patients” can be made based on just loose association, I recommend removing the sentence as it is not based on the study findings
9. Authors recommend testing for acquired vWF disease in cholestatic Fontan patients. How feasible is it in real practice? How often should the levels be tested? Is there a clinical significance of testing for mild avWF disease?
12. Why do authors repeat the subtitle "ENdothelial damage analysis" twice in the material and methods section? is that a typo?
10. I think the strength of this study is mainly in its negative findings: there is no statistically significant difference between all the markers of tissue factor activation, endothelial damage markers etc between cholestatic and non-cholestatic disease, it could be highlighted
11. I recommend adding more patients and redoing the analysis. With one group having only 6 patients, is it truly safe to draw any conclusions?
Author Response
Reviewer 1
I read the manuscript “Mild acquired von Willebrand syndrome and cholestasis in pediatric and adult patients with Fontan circulation” with great interest.
Here are my comments:
- Manuscript needs to be extensively revised for English and grammatical errors.
We revised the manuscript thoroughly for English and grammatical errors.
- Where did the cut-off of 10 micromole for bile acids come from?
The cut-off was established during validation of the HPLC-MS method in our laboratory with a broad range of samples from patients that were externally diagnosed with varying degrees or absence of cholestasis.
- How was the determination made between prescribing patients PhC vs. ASA?
The superiority of either anticoagulation or antiplatelet therapy in Fontan patients has not been proven yet. Therefore, treatment recommendation for each patient was based on subjective thromboembolic risk assessment of the treating physician. Generally, we are preferring the prescription of ASA for thrombophylaxis in our pediatric patients due to the higher risk of bleeding in case of trauma. At the appearance of clear indications such as atrial arrythmias or thromboembolic events, amongst others, pediatric and adult patients are switched to anticoagulation with PhC. Moreover, life events such as pregnancy and additional surgery might require changes to thrombophylaxis strategies.
- Was this a truly prospective study? Meaning were the patients enrolled in the study at the time when they underwent their Fontan with active consent about this specific study? If not, this cannot be considered a prospective study.
We agree with the reviewer and changed the designation to “observational study” (page 3, line 75).
- For TCPC, what was the approach? Lateral tunnel or atrial Fontans?
All included Fontan patients had a TCPC with lateral tunnel. We added this information in Table 1.
- Does the ventricular morphology in table 1 mean the dominant ventricle?
Yes, ventricular morphology stands for the dominant ventricle. However, we changed the description to “dominant ventricle” based on the reviewer’s valuable comment (Table 1).
- Figure 1a and 2 seem to be confusing. It may be better to place a proper Box plot instead of scattered dots.
We thank the reviewer for this valuable input. We now superimpose our data with boxplots to better depict the statistical distribution (Figure 1-3). However, we want to keep the information of individual patients in the plots, because they mark those patients with cholestasis (red).
- I do not think the conclusion of “ASA intake may reduce the risk 310 of advanced liver fibrosis in Fontan patients” can be made based on just loose association, I recommend removing the sentence as it is not based on the study findings.
We removed the sentence in the conclusion according to the reviewer’s recommendation (page 10, line 310).
- Authors recommend testing for acquired vWF disease in cholestatic Fontan patients. How feasible is it in real practice? How often should the levels be tested? Is there a clinical significance of testing for mild avWF disease?
Based on our findings, we are not able to make any recommendations concerning the frequency or the clinical meaning of mild AVWS in cholestatic Fontan patients. In this regard, ongoing clinical trials are required investigating the risk for bleeding in Fontan patients with signs of mild AVWS. Nevertheless, as the incidence rate is especially higher in cholestatic Fontan patients, it should be considered to test particularly those patients prior to surgery to avoid bleeding events. We rephrased our conclusion based on the reviewers comment to avoid confusion.
- Why do authors repeat the subtitle "ENdothelial damage analysis" twice in the material and methods section? Is that a typo?
We thank the reviewer for thoroughly reading of our manuscript and corrected the typo (page 3, line 105).
- I think the strength of this study is mainly in its negative findings: there is no statistically significant difference between all the markers of tissue factor activation, endothelial damage markers etc. between cholestatic and non-cholestatic disease, it could be highlighted.
We thank the reviewer for the valuable comment and highlighted the negative findings of our study in the conclusion.
- I recommend adding more patients and redoing the analysis. With one group having only 6 patients, is it truly safe to draw any conclusions?
We agree with the reviewer as stated in the limitations (page 10, line 299). Small case numbers are a general restriction when studying Fontan patients. However, the low incidence of cholestatic patients (based on serum total bile acid levels) is a novel clinical finding in Fontan patients and a key message of our study.

Reviewer 2 Report
The article titled „Mild acquired von Willebrand syndrome and cholestasis in pediatric and adult patients with Fontan circulation” embarks on analysis of very complexed relations between common in the study group liver failure and dangerous and also frequent coagulation disorders. Keeping in mind known from human physiology relationship between the accumulation of bile acids and hemostatic factors in patients after Fontan procedure, the Authors examined 34 patients with Fontan circulation and distinguished two study groups in regard to cholestasis presence. Evaluation of coagulation factors activity was related to aspirin administration and indirectly related to liver fibrosis.
This paper was performed by researchers specialising in pediatric cardiology, congenital heart defects in adults and hemathology and was planned and conducted tidely. This manuscript is methodologically correct.
However, under reviewer’s duty I present two remarks:
1. As an expert in congenital heart defects in adults I know that population of adults after Fontan procedure is small. Nevertheless the conclusions made by the Authors are too far-reaching. One can not draw conclusions based on analysis of 6 patients who were given aspirin. The effect of aspirin use would have tremendous prognostic implications but the study results do not entitle to such conclusions. Only suggestion of such possibility is justified and encouragement to further analysis. So the results should be more general and “soft”.
2. Reading this manuscript full of complexed abbreviations is very difficult. I know, this action enables inclusion of more data, nevertheless I would suggest withdrawal of some of them. For sure, the abbreviations should not be included in the conclusions.
Author Response
Reviewer 2:
The article titled „Mild acquired von Willebrand syndrome and cholestasis in pediatric and adult patients with Fontan circulation” embarks on analysis of very complexed relations between common in the study group liver failure and dangerous and also frequent coagulation disorders. Keeping in mind known from human physiology relationship between the accumulation of bile acids and hemostatic factors in patients after Fontan procedure, the Authors examined 34 patients with Fontan circulation and distinguished two study groups in regard to cholestasis presence. Evaluation of coagulation factors activity was related to aspirin administration and indirectly related to liver fibrosis.
This paper was performed by researchers specialising in pediatric cardiology, congenital heart defects in adults and haematology and was planned and conducted tidely. This manuscript is methodologically correct.
However, under reviewer’s duty I present two remarks:
- As an expert in congenital heart defects in adults I know that population of adults after Fontan procedure is small. Nevertheless the conclusions made by the Authors are too far-reaching. One cannot draw conclusions based on analysis of 6 patients who were given aspirin. The effect of aspirin use would have tremendous prognostic implications but the study results do not entitle to such conclusions. Only suggestion of such possibility is justified and encouragement to further analysis. So the results should be more general and “soft”.
We agree with the reviewer and have defused the statements in our conclusion (page 10).
- Reading this manuscript full of complexed abbreviations is very difficult. I know, this action enables inclusion of more data, nevertheless I would suggest withdrawal of some of them. For sure, the abbreviations should not be included in the conclusions.
We agree with the reviewer and have diminished the number of abbreviations. Moreover, we have excluded all abbreviations from the conclusion (page 10).

Round 2
Reviewer 1 Report
I thank the authors for making changes according to my suggestions.
Only thing I think remains is to add the answers to my questions 2 and 3 in manuscript to make the method more clear.
I congratulate the authors for their work.
Author Response
I thank the authors for making changes according to my suggestions.
Only thing I think remains is to add the answers to my questions 2 and 3 in manuscript to make the method more clear.
We added the answers to question 2 (page and 3, line 99)and 3 (page 9, line 259) in the manuscript.
I congratulate the authors for their work.
Thank you! All the best.